# Achieving an Optimal Fat Loss Phase in Resistance-Trained Athletes: A Narrative Review

**DOI:** 10.3390/nu13093255

**Published:** 2021-09-18

**Authors:** Carlos Ruiz-Castellano, Sergio Espinar, Carlos Contreras, Fernando Mata, Alan A. Aragon, José Miguel Martínez-Sanz

**Affiliations:** 1Faculty of Health Sciences, University of Alicante, 03690 Alicante, Spain; carlosruiznutricion@gmail.com; 2Faculty of Health Sciences, UCAM Universidad Católica de Murcia, 30107 Murcia, Spain; cjcontreras@ucam.edu; 3Centro de Estudios Avanzados en Nutrición (CEAN), 14010 Córdoba, Spain; fmataor@gmail.com; 4Department of Family and Consumer Sciences, California State University, Northridge, CA 91330, USA; alaneats@gmail.com; 5Research Group on Food and Nutrition (ALINUT), Nursing Department, Faculty of Health Sciences, University of Alicante, 03690 Alicante, Spain; josemiguel.ms@ua.es

**Keywords:** resistance training, bodybuilding, weight loss, fat loss, body composition, diet

## Abstract

Managing the body composition of athletes is a common practice in the field of sports nutrition. The loss of body weight (BW) in resistance-trained athletes is mainly conducted for aesthetic reasons (bodybuilding) or performance (powerlifting or weightlifting). The aim of this review is to provide dietary–nutritional strategies for the loss of fat mass in resistance-trained athletes. During the weight loss phase, the goal is to reduce the fat mass by maximizing the retention of fat-free mass. In this narrative review, the scientific literature is evaluated, and dietary–nutritional and supplementation recommendations for the weight loss phase of resistance-trained athletes are provided. Caloric intake should be set based on a target BW loss of 0.5–1.0%/week to maximize fat-free mass retention. Protein intake (2.2–3.0 g/kgBW/day) should be distributed throughout the day (3–6 meals), ensuring in each meal an adequate amount of protein (0.40–0.55 g/kgBW/meal) and including a meal within 2–3 h before and after training. Carbohydrate intake should be adapted to the level of activity of the athlete in order to training performance (2–5 g/kgBW/day). Caffeine (3–6 mg/kgBW/day) and creatine monohydrate (3–5 g/day) could be incorporated into the athlete’s diet due to their ergogenic effects in relation to resistance training. The intake of micronutrients complexes should be limited to special situations in which there is a real deficiency, and the athlete cannot consume through their diet.

## 1. Introduction

In sports nutrition, management of the body composition of athletes is a common practice. Loss of body weight (BW) in athletes is generally motivated by the desire to optimize performance by increasing the strength-to-body weight ratio (e.g., powerlifting or weightlifting) or to compete in a discipline limited by BW category (e.g., boxing), or for aesthetic reasons in sports that require it (e.g., bodybuilding) [1,2,3,4]. However, a hypocaloric diet could result in a significant loss of fat-free mass (FFM), which could affect the athlete’s performance [1,5,6]. Therefore, nutritional strategies during a weight loss phase in athletes, in addition to reducing fat mass (FM), should aim to maintain FFM [5,7,8].

Current recommendations regarding BW loss in athletes favor more moderate approaches with the intention of minimizing the negative effects associated with rapid BW loss. It has been observed that a loss of BW of 0.5–1.0 %/week, accompanied by a high protein intake and resistance exercises, could favor the retention of FFM during fat loss phases [1,3,4,5,7,8,9,10,11,12,13,14]. Nevertheless, the scientific literature on this topic in resistance-trained athletes is very limited. The latest research focuses on bodybuilders, being necessary scientific literature on dietary–nutritional and supplementation recommendations for fat loss in these athletes [4,15,16].

Unlike in other sports where strength exercise is used to complement to the athlete’s specific training, in disciplines such as powerlifting, weightlifting and bodybuilding, resistance training forms the basis of the training [17]. While in weightlifting and powerlifting, a reduction in FM could improve performance through an increase in the strength-to-body weight ratio [8,17,18], in preparation for a bodybuilding competition, a drastic reduction in FM is required to achieve optimal muscle definition [4,19,20,21,22].

The aim of this review is to provide a comprehensive overview of dietary–nutritional strategies for the loss of FM and the maintenance of FFM in resistance-trained athletes from a theoretical and contextual point of view, to enable focused future systematic reviews in some subtopics. This review can be an evidence-based guide for implementing the limited and relevant available data to resistance-trained athletes during periods of calorie restriction. 

## 2. Materials and Methods

The work comprised a descriptive study, involving a narrative review, to answer the research question, “What are the dietary–nutritional recommendations for optimal fat loss in resistance-trained athletes?”. A structured search was carried out in the PubMed, Epitesmonikos and Scopus databases, using descriptors as the Medical Subjects Headings (MeSH), entry terms and natural vocabulary related to the aim of the study (Appendix A). In addition, reference lists were used for further search of the selected papers for related references.

## 3. Energy Intake

To achieve weight loss, the implementation of an energy deficit is required, by reduction of the energy intake (EI) and/or an increase in caloric expenditure. The exercise activity thermogenesis (EAT) is one of the components of energy expenditure associated with physical activity-related energy expenditure [23,24]. The magnitude and duration of this energy deficit will determine the amount of weight loss [25]. The performance of resistance exercise prevents the loss of FFM during periods of caloric restriction (CR) [8]; however, resistance-trained athletes represent a population that already performs this type of training. Traditionally, the 3500 kcal [26,27] rule has been used, which postulates that after accumulating a weekly caloric deficit of 3500 kcal, 1 lb of BW (0.45 kg) is lost. However, this static mathematical model of weight loss has been repeatedly questioned since, among other issues, it does not necessarily account for the metabolic adaptations caused by the energy deficit itself [23,28,29], nor the non-lipid fraction of the adipose tissue [30,31]. Hall et al. [32] recently proposed the “rule of 55 kcal/day per pound of BW”. Hall et al. [30] explain that a person who maintains a deficit of 500 kcal per day could reduce their weight by 9 lb (4 kg) in total over a year (500 kcal/day divided by 55 kcal/day/lb), reaching a plateau at 12 months. With this 55 kcal/day rule, Hall et al. indicate a dynamic relationship between diet calories and weight loss and presents a more realistic view of the challenges experienced by patients with obesity [30]. However, after an objective analysis, it can be seen that the caloric intake of these people increased in an unconscious manner over time [33], due in part to the increased appetite experienced after the application of an energy deficit [34]. This fact, added to the physiological adaptations that reduce daily energy expenditure [35], facilitates the appearance of the so-called plateau in weight loss. Some of these adaptations are highlighted in the review of Trexler et al. [36], who concluded that after the implementation of a caloric deficit in order to lose fat, the body activates different mechanisms to minimize this weight loss. These include a reduction in daily energy expenditure (mainly due to the loss of body mass itself and the decrease in energy expenditure associated with physical activities other than exercise, the so-called non-exercise activity thermogenesis or NEAT), greater mitochondrial efficiency in the use of energy and changes in circulating hormone levels. Therefore, these factors should be considered, since, as the fat loss phase progresses, a lower caloric intake will be required to compensate for these adaptations. In aesthetic athletes, these adaptations are reduced, in part because these athletes are characterized by strict adherence to their diets [37], and by increased daily energy expenditure through cardiovascular exercise throughout contest preparation [38,39,40,41,42,43,44].

An important aspect to consider when determining the magnitude of the caloric deficit is the potential for more aggressive energy restriction to result in greater loss of FFM [1], as indicated by the results of several case studies which showed greater retention of FFM with weekly weight losses of 0.5% [45] than with losses of 0.7% [38] or 1% of BW per week [39]. In the case of the latter study, the loss of FFM amounted to 42.8% of the total weight. These data are in line with the last two revisions of the nutritional recommendations for athletes during their preparation for a natural bodybuilding competition [4,15], for which slow weight losses were recommended in order to preserve the maximum amount of FFM, especially in the final stage of preparation, since, as the competitor reduces his or her FM, the risk of FFM loss increases when he or she is subjected to a caloric deficit [8].

### 3.1. Low Energy Availability

Energy availability (EA) is a scientific concept that describes how much energy is available for the basic metabolic functions of the body when EAT is subtracted from EI [22]. To calculate EA, EAT is subtracted from EI and the result is divided by the FFM [ (EI-EAT)/FFM].

Low energy availability (LEA) consists of a mismatch between EI and EAT, resulting in an amount of energy that is inadequate to support the body’s requirements for maintenance of optimal health and performance. In women, EA for optimal physiological function amounts to 45 kcal/kg FFM/day [46]. Below a threshold of 30 kcal/kg FFM/day, hormonal alterations can appear (disruption in female sex hormones, decreases in T3, insulin, GH, IGF-1, leptin, and glucose, and increased cortisol). However, it has been observed that not all women respond in the same way to the same energy insufficiency [22,46,47]. Recently, Alwan et al. [20] reviewed the physiological and psychological implications of preparations for aesthetic competitions in female athletes, concluding that in these disciplines it is common for female athletes to remain below this threshold for prolonged periods of time, especially in the pre-competitive phase, in which energy availability ranged between 18.2 and 31.1 kcal/kg FFM/day [40,44]. Due to this, hormonal alterations and irregularities in menstruation [20] are quite frequent, and in some cases up to 71 weeks post-competition can be necessary to restore the menstrual cycle [40]. The appearance of irregularities in the menstrual cycle is frequently used as a secondary indicator of a long-term LEA; however, the use of contraceptive hormones could prevent this relationship from being established in some cases [20,48]. Considering the high prevalence of the use of hormonal contraceptives [20,49], it is possible that many female athletes who experience LEA are not detected, since the use of these contraceptives could maintain regular menstruation [20].

Fagerberg [22] reviewed the consequences of LEA in natural bodybuilders, concluding that, in men, an EA <25 kcal/kg FFM/day results in a greater loss of FFM, hormonal alterations, psychological problems, and, in cases where FM percentages close to 4–5% are reached, possible problems in the cardiovascular system [22]. It seems that the male reproductive system and the related hormones have lower energy requirements and, consequently, are affected to a lesser degree by LEA compared to the reproductive system in women [22,50,51].

### 3.2. Diet Break

A strategy proposed for the prevention of these adaptations is the implementation of a phase of time (1–2 weeks) in which caloric intake is increased to maintenance levels. Byrne et al. [52], in the so-called “MATADOR study”, implemented this intermittent caloric restriction strategy. The control group (CON) performed 16 consecutive weeks of CR while the intervention group (INT) performed blocks of 2 weeks of CR intercalated with 2 weeks of maintenance, repeating these blocks until also completing 16 weeks of CR, thus prolonging the intervention time to 30 weeks (16 for CR and 14 for maintenance). After 16 weeks of CR, the INT group had lost more BW, more FM and a similar amount of FFM. During the maintenance periods, this weight was not recovered, and in terms of the reduction in resting metabolic rate (RMR), there were no differences between the groups. However, when the results were adjusted for body composition (FM and FFM), the RMR reduction in the INT group was significantly lower (−86 vs. −179 kcal/day in the INT and CON groups, respectively). Nonetheless, this lesser reduction in the RMR group is not enough to explain the greater loss of fat experienced, and it is possible that in this group there was greater adherence to the diet prescribed by the researchers [52].

These results must be interpreted with caution as they derive from a sample of individuals with obesity who were physically inactive and did not perform resistance exercise; however, they can be used to generate hypotheses and new interventions. More recently, Peos et al. [53], comprising an intervention in strength athletes, the control group performed a linear CR for 12 weeks with 3 weeks of subsequent maintenance, whereas the intervention group performed 4 blocks of 3 weeks of CR with a maintenance week between each one. In addition to changes in body composition, the authors analyzed performance, physical activity, sleep and hormones involved in regulating hunger, among other variables. Peos et al. [53] concluded that similar fat loss and fat-free mass retention are achieved with linear CR and intermittent CR. For more information on this type of protocol, the recent review by Peos et al. [54] is recommended. Currently, the minimum number of days necessary to reverse the adaptations caused by CR itself and the frequency with which it would be necessary to increase energy intake until maintenance level are unknown. So, a “diet break” of two weeks could be excessively long, thus increasing the time necessary to achieve the desired fat loss, time that could be used to gain muscle mass [16].

## 4. Macronutrients

### 4.1. Protein

One of the main goals during a fat loss phase in strength athletes, in addition to reducing FM, is to preserve FFM. During sustained periods of energy deficit, the rate of muscle protein synthesis (MPS) is reduced in periods of fasting, after food intake and in the post-training period [55,56,57]. In addition, as muscle is a reservoir of amino acids, it can be catabolized during a period of negative energy balance in order to provide precursors for gluconeogenesis and oxidative energy metabolism [58,59], resulting in a negative net protein balance and a possible loss of musculoskeletal mass [55].

Morton et al. [60], in their recent systematic review, concluded that the protein requirements of athletes who perform resistance training are around 1.6 g/kg BW/day under energy sufficiency—that is, when there is a balance between the energy consumed and the energy required. However, the preparation phase prior to a competition is characterized by a period in which a sustained energy deficit is established and prevails over time, and several authors have suggested the need to increase protein intake during this stage [8,12,59,61,62]. Therefore, the optimal protein intake for resistance-trained athletes during an FM loss phase could be higher than the existing recommendations for optimization of muscle mass gains with adequate energy availability [60], as suggested in a recent opinion article [62].

Helms et al. [12], in the only systematic review of results obtained for athletes who perform resistance training during periods of CR, suggested a protein intake of 2.3–3.1 g/kg FFM/day. However, the conclusions of these authors should be viewed with caution due to the heterogeneity of the designs and samples of the studies reviewed [63]. Hector and Phillips [8], in a recent review of the protein recommendations for weight loss in elite athletes, suggested an intake of 1.6–2.4 g/kg BW/day, opting for the higher values in this range when the caloric deficit used is higher and the athlete’s body fat percentage is lower. This last aspect deserves special attention in the case of bodybuilders during competitive phases, since they reach extremely low values of body fat [37], a condition that makes them more susceptible to loss of FFM during an energy deficit [64,65]. The International Society of Sports Nutrition (ISSN), in its latest position on protein and exercise [61], suggests that trained strength athletes could benefit from diets high in protein; however, better results were not obtained with intakes greater than 2.6 g protein/kg BW/day. Recently, Bandegan et al. analyzed the protein requirements of bodybuilders on non-training days, using the indicator amino acid oxidation (IAAO) technique, and estimated needs of 2.2 g/kg BW/day [66]. It should be noted that these results were obtained 48 h after the last training session in young athletes (on average, 22.5 years old) who were following a diet adjusted to their caloric requirements. Therefore, these data must be interpreted with caution since we do not know if they are representative or can be extrapolated to training days, CR phases and/or older athletes [8,60,67,68]. An additional caveat is that the IAAO technique assesses protein synthesis at the whole-body level [69], which limits this method’s ability to draw firm conclusions about skeletal muscle anabolism specifically.

Another reason high-protein diets are proposed for a fat loss phase is their effect on satiety [70,71]. Dhillon et al. published a recent meta-analysis in which they concluded that high-protein meals increased the feeling of fullness to a greater degree than meals lower in protein [72]. However, this review focused on the short-term impact (up to 10 h) in untrained subjects and the results analyzed may not be extrapolatable to long-term satiety. In addition, it has not been determined whether, in a population that has previously consumed a diet high in protein, as is the case of bodybuilders or strength athletes [37], increasing the amount of protein consumed generates greater satiety. In relation to this issue, Roberts et al. performed a randomized, controlled trial in which strength-trained athletes underwent a short period of energy deficit [2]. The 16 participants were randomized into two groups, moderate protein intake (1.8 g/kg/ BWday) and high protein intake (2.9 g/kg/ BWday) for 7 days, matching the caloric deficit of both through a crossover design. The researchers concluded that there do not seem to be any benefits regarding perceived satiety when consuming high-protein diets, relative to diets with a moderate protein content [2]. Caution should be applied to this conclusion since the physical properties of food are known to affect satiety, and a liquid form (whey) was used to supplement protein intake in the high-protein condition. Since solid and more viscous foods tend to provide greater satiation and appetite suppression than liquid forms [73], questions remain about how results may have differed if solid protein sources were used. 

### 4.2. Carbohydrates

In sports nutrition, in terms of performance, the importance of carbohydrates (CHO), before, during and after intense and high-volume exercises, has been repeatedly reviewed [74,75,76], highlighting the dependence that exists in relation to their consumption in adequate amounts by athletes with competitive objectives, both in endurance disciplines and in team sports [77]. However, the role of CHO, and its manipulation in strength and aesthetic sports, has not been studied as widely [78,79,80].

Unlike endurance sports, in which a single multi-hour training session could completely deplete the glycogen stores [81], a strength training session can result in a 24–40% reduction in muscle glycogen stores [82,83,84,85], depending on the volume and intensity of the session, the musculature recruited and the rest between series [17]. The depletion is greater in those sessions in which exercises are performed with a high number of repetitions and moderate load [83]. This difference in the use of energy resources is important in order to set the CHO requirements for strength exercises; above all, to establish the minimum requirements below which performance could be compromised. On this issue, Slater and Phillips published a review of the nutritional requirements in strength sports, in which they concluded that a CHO intake of 4–7 g/kg BW/day would be adequate in strength athletes, according to their training phase [17]. Nonetheless, these conclusions should be taken with caution since they were elaborated solely on the basis of two aspects: (1) the CHO recommendation obtained as an average of the data extracted from multiple observational studies based on dietary surveys, which provided a consumption of 3–5 g CHO/kg BW in strength athletes and 4–7 g CHO/kg BW in bodybuilders [17], with the intrinsic limitations of this evaluation method [86]; (2) the Lambert and Flynn recommendation of 6 g CHO/kg BW for strength athletes [87], an amount based on two trials: one with glycogen depletion using a cycle ergometer [88] and another in which glycogen depletion using a cycle ergometer was combined with an eccentric training session with loads [89]. The latter is known to produce greater muscle damage and lower glycogen storage, among other factors, by increasing resistance to insulin [90]. The results obtained through these methodologies may not be representative of the real CHO requirements of resistance-trained athletes. However, in a recent review on CHO in strength and aesthetic sports published by Cholewa et al. [84], these CHO levels of 4–7 g CHO/kg BW recommended by Slater and Phillips [17] continue to be recommended for resistance-trained athletes. Despite the appearance of a consensus agreement on 4–7 g/kg BW, three recent reviews of CHO requirements in strength sports question this figure and show that CHO intakes of 1–3 g/kg BW may not reduce performance or interfere with post-exercise cell signaling [78,79,80]. On the other hand, Chappell et al. [21] observed that natural bodybuilders who placed in the top five in competitions had higher CHO consumption before and during the preparation for competition than those who placed out of the top five. From this, the authors concluded that the higher intake of CHO could have contributed to the maintenance of FFM during preparation. However, the mean number of years of training of those classified in the top five was 3.3 years higher than for the unclassified athletes (14.2 and 10.9 years, respectively), and the energy intake at the beginning, in the intermediate phase and at the end of the competition was ≈400 kcal/day higher in the classified group, so these results should be interpreted with caution.

The CHO consumption during the FM loss phase in resistance-trained athletes could be established in a wide range (2–5 g/kg BW), adjusting the intake to the individual caloric requirements of the athlete and their food preferences, in order to increase the adherence to the diet. This wide range is mainly due to the low reduction in muscle glycogen during a strength session [82,83,84,85], which reduces the requirements compared to other disciplines [77]. However, although low-CHO diets can be effective, there is a muscle glycogen threshold below which performance during training may be affected if muscle contraction is compromised. When muscle glycogen falls below 70 mmol/kg dry weight, the release of calcium from the sarcoplasmic reticulum is impaired, as is the maximum power [81], which indicates that there is a threshold for muscle glycogen below which muscle function and performance could be affected [81]. Therefore, the minimum CHO intake should ensure a replenishment of muscle glycogen so that it at least exceeds this lower limit (70 mmol/kg) during training, this amount (expressed in g/kg BW/day) being variable based on the athlete’s own characteristics, as well as the rest of the activities carried out in addition to training (e.g., physically demanding jobs). 

With that said, it should be noted that the goal of increasing or preserving maximal strength may allow greater flexibility with lower intakes of CHO. The majority of investigations of the effect of ketogenic diets on resistance training performance have not shown decrements compared to control/non-ketogenic conditions [91]. This preservation of performance (lifting strength) following ketogenic diets has been demonstrated in resistance-trained subjects at moderate [92] and near-maximal to maximal loading [18,93,94]. Therefore, while ketogenic diets (<10% of total energy by CHO, or ≤50 g CHO/day) might be sub-optimal for maximizing FFM preservation [18,93], they do not appear to have the same potential for compromising strength preservation.

To avoid the ergolytic potential of low glycogen storage, in addition to opting for a diet with a higher daily CHO content, the strategy known as “Refeed” or “carbohydrate loading” can also be used. This consists of increasing dietary CHO and calories to levels equal to or higher than maintenance in a timely and scheduled manner within the planning [36]. Generally, in bodybuilding, the protocols used last 24 h, once or twice a week [95], although the trials that have studied refeed protocols used them for at least three days per week [96,97]. The supposed objective of this strategy is to temporarily increase the circulating leptin levels and stimulate the metabolic rate [36]. There is evidence that leptin is sensitive to brief periods of refeeding with CHO, but not with fats (FAT) [96]. Dirlewanger et al. observed that, after three days of refeeding with CHO, leptin had increased by 28% and daily energy expenditure by 7%. However, this protocol consisted of a caloric overingestion of 40% above maintenance for three days to achieve only a 7% increase in energy expenditure [96], which returns to baseline values once the caloric deficit is restored [98,99]. This increase in RMR has been verified in athletes in post-competitive periods. For example, Trexler et al. observed how energy expenditure was higher in the weeks after a competition than in the days before; in these weeks, the intake of calories, CHO and fat was higher [100]. This proposed increase in energy intake to maintenance levels by increasing CHO should be offset by a reduction in energy intake on the other days of the week in order to maintain the programmed caloric deficit.

To conclude the CHO section, the recommendations established by European Food Safety Authority (EFSA) for fiber consumption are 25 g/day [101], an amount applicable to the sports population. An increase in fiber could be useful in the search for greater satiety during the CR phase [102], although its excessive consumption could be detrimental to the absorption of some nutrients [103]. 

### 4.3. Fats

In sports nutrition, the intake of FAT should, as in the general population, facilitate adequate consumption of essential fatty acids and fat-soluble vitamins, replenish intramuscular triglyceride stores, and maintain the energy balance [77,104]. The manipulation of this macronutrient, regarding the amount and source of dietary FAT, could have an impact on health and on the concentration in the blood of some anabolic hormones, which could also affect body composition and performance [4,77,104,105,106,107]. However, this effect on anabolic hormones could be due to the limited calorie intake—so, to draw conclusions, more studies are needed on this topic.

Aesthetic athletes, for example, are required to undergo CR periods in their respective pre-competitive phases to achieve the desired physique. Some authors have observed that diets with a low fat intake (≤20% FAT) can reduce testosterone levels [106,108]. Nevertheless, it is difficult to extract a direct association of these two variables due to other characteristics of the trials; in addition to a low fat intake, the subjects undergoing CR had a low percentage of body fat and a low intake of saturated fat and polyunsaturated fatty acids [4,107,109,110,111]. If, to establish the energy deficit, it is decided to reduce the contribution of FAT, the recommendation is to ensure an intake of 20–30% of the total daily energy supply or, if that is not possible due to the caloric limitation and to prioritization of an adequate intake of PRO and CHO, a daily FAT intake of at least 0.5 g/kg BW should be ensured [77]. Regarding the reduction in testosterone levels with low fat intakes, increasing the intake of saturated fat could improve the situation [4,112], as could opting for less aggressive weekly weight losses (0.5% BW/week) [113].

In terms of maintenance of FFM, low-calorie diets high in PRO, high in CHO, and low in FAT [1,114] yielded better results than those interventions that opted for high PRO and FAT, but a low contribution of CHO [55,115]; to the latter studies we can add the data obtained recently by Chappell and collaborators, mentioned previously [21].

In aesthetic sports, due to the characteristics of these disciplines, sometimes it will not be possible to consume the recommended intake of FAT [77]—for instance, when energy requirements are limited and an adequate consumption of PRO and CHO is prioritized.

## 5. Nutritional Timing

For the International Society of Sports Nutrition (ISSN), “nutritional timing encompasses the intentional intake of all kinds of nutrients at various times throughout the day that have a positive impact on the acute and chronic response to exercise” [116].

Most of the research related to nutrient timing is based on endurance sports, with results and recommendations being directly extrapolated to strength and aesthetic sports on several occasions [78,79,80]. One of the representative examples of this erroneous extrapolation of the results is the recommendation of a rapid intake of CHO at the end of the training session in order to rapidly replenish muscle glycogen. In endurance sports, where high volumes of training sessions can significantly reduce or even completely deplete glycogen stores [81], it is justifiable to take advantage of the greater glycogen resynthesis capacity that occurs in the hours after training [116]. However, the need for this rapid glycogen replacement has not been demonstrated in resistance trained athletes, who perform only one training session per day. If extra sessions of cardiovascular exercise are added in order to increase the total daily energy expenditure (for example, for bodybuilders during contest preparation), it might be advisable to include CHO intake at the end of the first training session, to replenish glycogen stores and prepare for the second session of the day.

Trials examining the timing of CHO intake in strength sports are scarce [116]. Those studies in which pre-strength training CHO were administered to subjects with moderate glycogen depletion found no improvement in performance [116]. To date, there is only one report of positive results in terms of performance when administering CHO before and during different series of resistance exercises [117]. However, ergogenic effects were only observed in the second resistance training sessions carried out on the same day [117]; therefore, prioritizing the fulfillment of total daily CHO requirements seems to be more important than the temporal aspects of consumption relative to the training sessions.

Regarding the protein intake by participants in aesthetic and strength sports, several important aspects should be taken into account. One of them is the time of intake (pre- vs post-training). Several reviews have observed that when the total daily protein amount is the same for both groups, no differences are obtained between pre- and post-training intake [118,119]. Another issue regarding protein timing is the intake of protein before sleep. Recently, Snijders et al. [120] extensively reviewed the impact of protein intake prior to sleep on the adaptive response to exercise of skeletal muscle, reaching the conclusion that this intake is an effective strategy to increase the rate of MPS during nighttime sleep and can be applied as a tool to benefit the adaptive response of skeletal muscle to resistance exercise [120]. Despite the conclusions reached by these authors, analysis of the studies that make up this review highlights an important confounding variable that does not appear in the conclusions, but does appear in the discussion. In those studies in which greater gains in hypertrophy and/or strength were achieved, as in the case of the results obtained by Snijders et al. [121], by providing a placebo to the non-supplemented group, the daily protein intake of the group with pre-sleep protein was superior to that of the control group: 1.9 vs 1.3 g/kg BW respectively. Therefore, the greater gains in strength and hypertrophy cannot be attributed solely to protein intake prior to sleep, since it has not been compared with a protein intake of the same magnitude at any other time of the day. It is more likely that the greater gains occurred because the supplemented group consumed an amount of protein close to that recommended to optimize muscle mass gains [60]. In fact, when the amount of protein is equalized, enabling a comparison of the same amount of protein administered at night or at another time of the day, the results are similar for both groups [122,123]. Therefore, based on the current evidence, pre-sleep protein intake could be recommended in those situations in which the recommended daily amount of protein is not reached, while remaining aware that the intake of this protein could occur at any other time of the day, as it would produce similar results.

Probably, the most important factor in protein timing is how the intake is distributed throughout the day. Regarding the amount of protein ingested per meal, there seems to be a minimum protein threshold (mainly dependent on the leucine content) required to stimulate MPS [124,125]: 0.40–0.55 g/kg BW, increasing to 0.60 g/kg BW in the elderly [126], this being the amount of protein per intake which maximizes the MPS response, since it ensures that in each of these intakes the leucine content is ≥1–3 g [61].

However, in recent years, the hypothesis of the “muscle full effect” has emerged, which states that the high presence of amino acids in plasma stimulates MPS for a short period of time, after which a refractory period appears in which the muscle seems not to respond to the increase in amino acids [127,128]. After protein intake, anddepending on the kinetics of the protein ingested, there is a period of 45–90 min before the MPS rises; it reaches a maximum at 90–120 min, before returning to baseline values even though the amino acid levels remain high [127]. This suggests that a 2–3 h period should be allowed between protein intakes to enable MPS re-stimulation. This theory requires more research since, at the moment, the mechanisms of action are not known [124], and there are results that contradict it, such as those of Churchward-Venne et al. [129]. These authors observed that, in the group that ingested 25 g of whey protein after training, the response of MPS was greater at 3–5 h than at 1–3 h post-ingestion, a situation that was not replicated in the group that did not perform strength training but did ingest the same amount of protein (128). This indicates that, at least post-exercise, the “muscle full effect” theory has limitations.

Regarding the co-ingestion of CHO and PRO, it has been established that when both are ingested in adequate amounts, their combination does not result in any improvement in performance or in short and/or long-term adaptations [116] to resistance training. 

## 6. Frequency

In strength sports, traditional dietary protocols have featured a high number of daily intakes [37,95]. The reasoning is usually that they: (1) increase metabolism and (2) maintain a constant supply of amino acids. The second point has already been discussed in the previous section, where it was mentioned that continued high levels of amino acids in the blood do not necessarily induce a greater or more extensive response of MPS [127,128]. Regarding the increase in metabolism, two studies involving metabolic chambers showed that no significant differences in thermogenesis were induced by changes in the frequency of ingestion, using a wide range (two to seven) of daily meals [130,131]. 

Currently, there is a trend towards a lower number of meals per day (for example, 1–2) restricted to a short period of time during the day. Recently, Peos et al. reviewed the available literature on intermittent caloric restriction, concluding that, in those studies in which energy and nutrient intake are equal, and the only difference is the number of meals per day or the time window of feeding, there are no differences in terms of body composition [54]. Trials regarding intermittent fasting or time-restricted feeding (TRF; typically involving an 8-h feeding window) and maintenance of FFM during periods of caloric deficit show mixed results, finding studies in which there is no difference in loss of FFM [132,133,134] and other trials in which there is less preservation of FFM in the group that performed intermittent fasting [135].

Therefore, once the most relevant aspects of protein timing and the frequency of daily meals have been considered, the recommendations regarding protein intake could be summarized as: 1. Consume an adequate daily amount (2.2–3.0 g/kg BW); 2. Select the eating pattern that generates the greatest adherence in the person, with 3–6 feedings in which the amount of protein is ≥0.4 g PRO/kg BW, using sources of high biological value, thus ensuring a leucine intake ≥2.5 g [4]; 3. Carry out one protein feeding within 2–3 h before training and another within 2–3 h after training [118]. 

## 7. Micronutrients

The available literature on the intake of micronutrients by resistance-trained athletes during a weight loss phase is scarce. In weightlifters, an intake below the recommended daily allowance (RDA) of vitamins B1, B3, B6, and B9 and magnesium [136], as well as a deficiency in the consumption of magnesium, calcium and potassium [137] has been described.

In 2014, Helms et al. [4] highlighted the nutritional deficiencies in the diets of bodybuilders, based on previous observational studies in which the amounts of vitamin D, calcium, zinc, magnesium and iron ingested did not reach the recommended values. However, the authors underlined the need for more studies since those available were more than three decades old and may not be representative of the current population of bodybuilders. Spendlove et al. [37] carried out a systematic review of the dietary intake of competitive bodybuilders; again, this was hampered by the fact that the only available studies were prior to the 1990s and most of them concerned only the consumption of vitamin supplements.

Of particular relevance to hypocaloric conditions, Calton [138] analyzed the micronutrient sufficiency of four popular diet programs by comparing their content of 27 essential micronutrients with reference daily intake (RDI) standards. A typical dieter on any of these four popular diet plans would be, on average, 56.48% short of RDI sufficiency, and lacking 15 of the 27 essential micronutrients analyzed. These, as well as the findings of other investigators [139] illuminate the high prevalence of incomplete micronutrient coverage among dieting populations. It is plausible that similar shortcomings can occur in resistance-trained athletes in hypocaloric conditions—especially in individuals with limited dietary diversity within and across the range of food groups.

In the absence of studies with large, representative samples of resistance trained athletes that reflect the actual and current consumption of micronutrients, the benefit of multivitamin-mineral (MVM) supplementation remains an unresolved issue, drawing a divided camp for and against its use. Arguments against vitamin supplementation hinge primarily upon their lack of clinical benefit. Nevertheless, Bird et al. [140] reported that 31% of the United States population is at risk of deficiency in at least one vitamin, or has anemia. Dietary supplement non-users had the highest risk of any deficiency (40%), while this risk in MVM consumers was substantially lower (14%). If a nutritional deficiency is suspected or detected, the first option should be to try to correct it by improving the nutritional pattern; if this is not possible, the deficient vitamin or mineral should be supplemented specifically in the appropriate amount. Nevertheless, routine or prophylactic use of low-dose MVM (most nutrients dosed at or near the RDI) is still worth considering, due to widespread subpar micronutrient intakes across various populations. The potential benefits of MVM should not be dismissed or ignored [141]. Even when a diet is well planned, it is still not always possible to meet all of the recommended intakes of the full range of essential micronutrients. In those cases of low energy intake, where even the intake of vegetables, fruits and other foods with high nutrient density may be reduced, the intake of multivitamins could alleviate potential inadequacies. A relatively recent review by Ward [142] is worth quoting directly: “When deciding whether to recommend the use of dietary supplements, it is important to consider the benefit:risk ratio. Current data suggest minimal, if any, risk associated with MVM preparations containing 10 or more vitamins and minerals at recommended daily intake levels in healthy people and a possibility of modest benefits that include a reduced risk of cancer and nuclear cataract, for a relatively low financial cost”. 

## 8. Supplementation

In a phase of fat loss in resistance-trained athletes, the fundamental pillars are training with loads, nutrition, rest and, if carried out, cardiovascular exercise, although some supplements could be useful. The objective of this section is not to carry out a review of all the supplements used by resistance-trained athletes; instead, it is focused on the usefulness of the two supplements with the most proven efficacy in the scientific literature: caffeine and creatine monohydrate. For information regarding other supplements with possible applicability during the fat loss stage, these recent reviews on supplementation are recommended [77,143,144]. Whey protein is not included in this section since its use is mainly as one more source of protein in the diet; for more information, the recent review of the ISSN regarding protein and exercise is recommended [61].

### 8.1. Creatine Monohydrate

Creatine is produced naturally in the body from the amino acids glycine, methionine and arginine, and is used in the phosphocreatine energy system in explosive activities lasting 0–10 s. It has been shown consistently that after administration of creatine supplementation, there is an increase in the intramuscular creatine concentration, which helps to explain the results obtained in terms of improved performance in high-intensity exercises, thus leading to greater adaptations to the training [145].

Most studies that have proven the efficacy of creatine as a supplement—regarding muscle phosphagen levels, body retention of creatine, and/or performance—have involved creatine monohydrate. However, creatine as a supplement is marketed in many other forms: creatine citrate, creatine ethyl ester, alkaline creatine, creatine nitrate, etc. [145]. For all of them it is claimed that there is lower degradation or greater retention at the muscular level, but this has not been demonstrated [145], and the bioavailability of some of these alternative forms is even lower [146]. So, it is recommended to choose the most studied and cheapest form, such as creatine monohydrate.

In the context of this review, the use of creatine to lose fat and maintain muscle mass relates mainly to its effects on the concentration of muscle phosphocreatine, increases in phosphocreatine resynthesis, the reduction of the muscle acidosis produced in high-intensity exercises, and muscle mass and strength gains [145]. This increase in phosphocreatine deposits would sharply increase the ability to exercise at high intensity and thus would generate greater adaptations, allowing more work to be done during the series and leading to greater gains in strength, muscle mass, and/or performance as training quality increases [145]. In addition, athletes may benefit from creatine supplementation indirectly, since it has been observed that creatine supplementation in combination with strength training could increase the training-induced proliferation of satellite cells and myonuclei in skeletal muscle, resulting in increased muscle fiber growth [147].

With a regular diet, creatine stores are at 60–80% saturation; therefore, creatine intake through supplementation would serve to increase muscle creatine and phosphocreatine levels by 20–40% [145]. There are several ways to reach these levels: for instance, using a 5-day loading phase or through maintenance intake, reaching these values in 28 days.

With a load: 5–7 days in a row with 0.30 g/kg BW of creatine monohydrate, divided into 4 intakes throughout the day. Once the muscle creatine levels have been saturated, the intake of creatine monohydrate is reduced to maintenance (3–5 g/day).

Without a load: daily intake of 3–5 g/day, up to 5–10 g/day in larger athletes of creatine monohydrate, reaching saturation of muscle creatine levels in 28 days; the time of day at which it is taken is not important in terms of the long-term results [145].

With regard to the benefits of creatine supplementation in the long term, neither the timing, nor its combination with CHO intake, nor the carry out of the loading phase are factors that will influence the improvement in sports performance. The loading phase would only be recommended in those cases in which saturation of phosphocreatine stores is required in a short period of time; for example, if it is decided to start the supplementation in the middle of the FM loss phase [145].

For a complete review of the literature regarding the use of creatine in different areas of performance and health, it is recommended to read the recent document prepared by the ISSN that details its position on the safety and efficacy of the use of creatine supplementation in exercise, sports, and medicine [145].

### 8.2. Caffeine

Together with beta-alanine, caffeine is the most common supplement in the 100 best-selling commercial pre-workout products, their prevalences being 87% and 86%, respectively [148]. Caffeine is a stimulant, which can be found in nutritional supplements, naturally (in foods such as coffee, tea and chocolate), or added to energy drinks and soft drinks. It has been shown to be effective as an ergogenic substance in both high- and moderate-intensity exercises, and its possible use in periods of caloric restriction (to increase energy expenditure) has been explored [77], although due to the magnitude of this increase it may not translate into a long-term significant fat loss.

This review is concerned with the action of caffeine, taken through supplementation or in coffee, as a stimulant, which does not seem to be affected in chronic coffee or caffeine consumers [149]. Regarding its effect on performance in strength sports, Grgic et al. published a recent systematic review analyzing the acute effects of caffeine ingestion on maximum strength and power, measured through the vertical jump, in which they came to the conclusion that caffeine intake is effective in improving muscle strength and power. However, in a later analysis, these authors observed that it was in the upper body where caffeine supplementation produced the most pronounced effects on performance, rather than in the lower body [150]. These results were replicated in a trial conducted by the same researchers [151]. The results obtained in terms of the ergogenic effect of caffeine appear to be similar when the same amount of caffeine is administered through coffee [150], although some aspects such as the large volume of liquid provided and the difficulties regarding quantification of the exact amount of caffeine ingested should be taken into account [152]. The recommended dose to reduce fatigue is 1–3 mg/kg BW/day [149] and to improve performance in strength training it is 3–6 mg/kg BW/day, 30–60 min prior to exercise [77,153,154]. However, recent studies show the great inter-individual variability in the response to different doses of caffeine, mediated by genetic and non-genetic factors [155]; therefore, both the dose and the timing should be individualized for each athlete.

On a cautionary note, caffeine has the potential to interfere with the ergogenic effects of creatine [156]. Possible mechanisms include gastrointestinal upset as well as opposing actions on muscle relaxation time, via opposing effects on calcium kinetics [157]. However, the concern of caffeine and creatine’s incompatibility has been challenged by ergogenic effects of creatine mixed with coffee [158] and tea [158], as well as the multi-ingredient supplements containing both caffeine and creatine, yet consistently improving exercise performance and increasing lean mass [159,160]. 

## 9. Conclusions and Practical Applications

Resistance-trained athletes undergoing a weight loss phase should focus their efforts on maximizing FFM retention while reducing the fat mass. The caloric intake should be set based on a target BW loss of 0.5–1.0%/week in order to maximize retention of FFM. The lower the % body fat of the athlete, the more conservative should the energy deficit be. The recommended protein intake is 2.2–3.0 g/kg BW/day, distributing this throughout the day in three–six meals and ensuring in each of them an adequate amount of protein (0.40–0.55 g/kg BW/intake). The carbohydrate intake should be adapted to the athlete’s activity level in order to promote performance during training (2–5 g/kg BW/day). Individuals who wish to engage in more severe CHO restriction (e.g., ketogenic conditions) may increase the risk of FFM loss, despite a similar capacity to preserve strength. Once the protein and carbohydrate intake has been established, the rest of the calories can be assigned to fat, ensuring a minimum intake of ≥0.5 g/kg BW/day. Regarding protein timing, an intake 2–3 h before training and another 2–3 h post-training is preferable. The intake of caffeine (3–6 mg/kg BW/day) and creatine monohydrate (3–5 g/day, up to 5–10 g/day in larger athletes) could be incorporated into the athlete’s diet due to the ergogenic effects related to resistance-training. Specific vitamin supplementation should be limited to special situations in which there is the detection of, or high risk for deficiency—and the athlete cannot consume the recommended daily amount of these nutrients through dietary sources. Routine MVM use remains controversial but its benefits likely outweigh its risks. The main limitation of this review is the small number of long-term studies with large samples conducted on resistance-trained athletes during a weight loss phase. More research is required in this population in order to expand our knowledge and improve nutritional and dietary supplement recommendations. Table 1 summarizes the conclusions of each subsection.

## Figures and Tables

**Table 1 nutrients-13-03255-t001:** Summary of dietary–nutritional recommendations for natural bodybuilding. Source: compiled by authors.

**Weight Loss**	0.5–1.0% BW/week. Opt for low-end range.
**Protein**	2.2–3.0 g/kg BW/day.
**Carbohydrates**	2–5 g/kg BW/day.
**Fats**	0.5–1.0 g/kg BW/day.
**Timing and Frequency**	Three–six meals/day. Four–five protein intakes (0.4–0.55 g/kg BW). One protein intake 2–3 h pre-training. One protein intake 2–3 h post-training.
**Micronutrients**	Include a variety of fruit and vegetables. In case of deficiency, the specific vitamin/mineral should be supplemented. MVM is worth considering, especially in hypocaloric conditions involving a limited spectrum of food diversity.
**Supplements**	Creatine: With load: 5–7 days with 0.30 g BW/day, divided into four doses, and continue with maintenance of 3–5 g/day. Without load: 3–5 g/day, up to 5–10 g/day in larger athletes. Caffeine: To reduce tiredness: 1–3 mg/kg BW/day. To improve performance: 3–6 mg/kg BW/day 30–60′ pre-training.

BW—body weight; EI—energy intake.

## Data Availability

The data presented in this study are available in the tables and Appendix A of this article.

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
