# Peer review of "Achieving an Optimal Fat Loss Phase in Resistance-Trained Athletes: A Narrative Review"

_nutrients, 2021, doi:10.3390/nu13093255_

Round 1

Reviewer 1 Report

Carlos Ruiz-Castellano et al. discussed the issue about achieving an optimal fat loss phase in resistance trained athletes. The conclusions are; Resistance trained athletes undergoing a weight loss phase should focus their efforts on maximizing fat-free mass retention while reducing the fat mass. The caloric intake should be set based on a target BW loss of 0.5-1%/week in order to maximize retention of fat-free mass. The lower the % body fat of the athlete, the more conservative should the energy deficit be. The recommended protein intake is 2.2-3 g/kg BW/day, distributing this throughout the day in 3-6 meals and ensuring in each of them an adequate amount of protein (0.4-0.55 g/kg/intake). The carbohydrate intake should be adapted to the athlete's activity level in order to promote performance during training (2-5 g/kg BW/day). Individuals who wish to engage in more severe CHO restriction (e.g., ketogenic conditions) may increase the risk FFM loss, despite a similar capacity to preserve strength. Once the protein and carbohydrate intake has been established, the rest of the calories can be assigned to fat, ensuring a minimum intake of ≥0.5 g/kg/day. Regarding protein timing, an intake 2-3 h before training and another 2-3 h post-training is preferable. The intake of caffeine (3-6 mg/kg BW/day) and creatine monohydrate (0.08-0.1 g/kg BW/day) could be incorporated into the athlete's diet due to the ergogenic effects related to resistance training. Specific vitamin supplementation should be limited to special situations in which there is the detection of, or high risk for deficiency - and the athlete cannot consume the recommended daily amount of these nutrients through dietary sources. Routine MVM use remains controversial, but its benefits likely outweigh its risks. The main limitation of this review is the small number of long-term studies with large samples conducted on resistance-trained athletes during a weight loss phase. More research is required in this population in order to expand our knowledge and improve nutritional and dietary supplement recommendations.

Major comments

  1. First of all, the paper is too long to be difficult to be understood. At least, the introduction and conclusions should be shortened. The conclusions are shown above, but what is the essence of this study?
  2. Second, it is uncertain what was done in this study based on the abstract and introduction. What was the hypothesis? What was the purpose of this study?
  3. In the title; the type of the study should be identified.
  4. The abstract is awkward; this reviewer can not understand what was done.
  5. What is the clinical relevance of this study?
  6. Clinical question should be set before writing this paper.
  7. At least, this paper should be useful for readers. The paper should be a systematic review and meta-analysis.

Specific comments

  1. Abstract: vitamin complexes need to be defined here.
  2. Abstract and test: 0.5%-1.0%/week, 2.2-3.0g/kg BW/day, and 0.08-0.10 g/kg BW/day may be adopted.
  3. Introduction: the aim of this review is used twice. So, the aims of this review should be appropriate.
  4. Energy intake: what is EAT? It should be defined precisely. Furthermore, this section is too long.
  5. 1. Diet break: this section is too long, which could be concise.
  6. 1. Protein: this section is too long, which could be concise.
  7. 2. Carbohydrates: this section is too long, which could be concise.
  8. Nutritional timing: this section is too long, which could be concise.
  9. Micronutrients: this section is too long, which could be concise.
  10. 1. Creatinine monohydrate: this section is too long, which could be concise.
  11. 2. Caffeine: this section is too long, which could be concise.
  12. 8 Conclusions: this section is too long, which could be concise.

Author Response

We attached the cover letter to reviewer 1

Reviewer 2 Report

This is a fairly good review of the topic but there are areas that can be improved upon. One major area is how this review expands on previous reviews and the novelty of the review. So there definitely needs to be a stronger rationale for the review. There are inconsistencies with the use of abbreviations so make sure you revise (e.g. fat-free mass and FFM). Additionally, when mentioning relative nutrient prescription make sure to include what the per kg is referring to (e.g. BW or FFM). More specific comments and feedback is provided below.

Line 26: “...promote performance during training…” -  Be more specific with what you mean here.

Line 49: “…to complement an athlete's…”

It would be good to provide some background on the previous reviews conducted on this subject (or similar) and discuss how what your review adds to the existing body of evidence.

Line 67: What does ‘EAT’ stand for?

Line 68: “…weight loss.”

Lines 75-78: Please provide a clearer explanation for this equation. So you are saying irrespective of the caloric deficit, 55kcal per LB of BW in 12 months? How about over 3 months etc.?

Line 79: “was people”? were a cohort comprising of participants that were…..?

Lines 78-82: It would be easily to follow if this sentence was divided into two sentences.

Lines 83-84: This sentence is difficult to understand. Please revise.

Line 91: “…changes in circulating hormone levels..”

Line 127: “usually used” – please revise.

Lines 136-138: You are talking about the reproductive systems however hormones are not part of this system? Please revise.

Line 141: “full periods (1-2 weeks)” – menstrual periods?

Lines 164-169: Are you referring to “refeeding” phases?

Line 175: Spell out “MPS” before abbreviating.

Line 184: “…characterized by a period..”

Line 226: Spell out “PYY” before abbreviating.

Lines 259-266: Please break up this extremely long sentence to at least two sentences.

Line 271: 4-7 g/kg/BW? Also in other places make sure that per kg of ?? is specified.

Line 291: “..below 70 mmol/kg dry weight..” Can you put this into a ‘real world’ perspective?

Line 303: “following ketogenic diets”

Line 305: “(<10% of total energy, or ≤50 g/day) – what are you referring to here, CHO? Please specify.

Line 327: Spell out “EFSA” before abbreviating.

Lines 424-427: Please revise this sentence so that it is easier to understand.

454-457: You are not abbreviating fat-free mass (FFM) here but have done so elsewhere. Please be consistent.

Line 510: “MVM consumers”

Line 579: What is meant by “nor the performance of the loading phase”?

624-627: You are not abbreviating fat-free mass (FFM) here but have done so elsewhere. Please be consistent.

Line 633: “risk of FFM loss”?

Table 1: Need to mention if per kg is per BW or FFM?

Also, what is “3-6 intakes/day” referring to?

Author Response

we attached the cover letter to reviewer 2

Round 2

Reviewer 1 Report

The comments and questions have been addressed.

Reviewer 2 Report

Well done on addressing my concerns.